# Behavior of Mebendazole during NF/RO Adsorption and Photolysis

**DOI:** 10.3390/membranes12090888

**Published:** 2022-09-15

**Authors:** Bruna Babić, Darko Andrić, Anamarija Farkaš, Dragana Vuk, Danijela Ašperger, Davor Dolar

**Affiliations:** 1Department of Analytical Chemistry, Faculty of Chemical Engineering and Technology, University of Zagreb, Marulićev trg 19, 10000 Zagreb, Croatia; 2The Institute for Development and International Relations, Ljudevita Farkaša Vukotinovića 2, 10000 Zagreb, Croatia; 3Department of Organic Chemistry, Faculty of Chemical Engineering and Technology, University of Zagreb, Marulićev trg 19, 10000 Zagreb, Croatia; 4Department of Physical Chemistry, Faculty of Chemical Engineering and Technology, University of Zagreb, Marulićev trg 19, 10000 Zagreb, Croatia

**Keywords:** mebendazole, reverse osmosis, retentate, adsorption, photolysis, toxicity

## Abstract

The idea of using drugs from the benzimidazole group as potential antitumor agents is becoming increasingly popular and widespread in research. However, their use as antiparasitics and in cancer treatment will increase their already recorded occurrence in the aquatic environment. In this study, the removal of the anthelmintic mebendazole from aqueous solution was investigated using nanofiltration and reverse osmosis membranes, adsorption on granular activated carbon (GAC), and photolytic degradation. The dense NF90 and reverse osmosis XLE membranes showed almost complete removal (>97.7%), while the NF270 membrane showed a large dependence of removal on initial concentration from 41.9% to 96.6%. Adsorption in the column resulted in complete removal of mebendazole at the highest GAC height used (40 cm) from the solution with the lowest concentration (1 mg/L). Photolytic degradation by artificial light for 2 and 12 h resulted in photodegradation of mebendazole in the range of 23.5–61.4%, forming a new degradation or transformation compound with an *m/z* ratio of 311. Mebendazole is a photosensitive drug whose photodegradation follows first-order kinetics and depends on the drug concentration. Toxicity was studied with *Vibrio fischeri* before and after photolysis, and showed a decrease in inhibition after 12 h.

## 1. Introduction

The problem of water scarcity as a result of increasing droughts, global industrial development, population growth, and pollution of available resources has attracted much attention. Conservation of limited drinking water resources is becoming one of the most important goals. As a result, there is growing interest in water recycling processes and in developing more efficient treatment processes. A major problem in conserving water resources and the overall balance of the ecosystem is pharmaceuticals, which cannot be adequately removed by conventional wastewater treatment processes.

Pharmaceuticals are no longer new pollutants, but due to their constant use, they represent a diffuse and pseudo-persistent environmental pollution whose interaction in the ecosystem is not yet fully explored. Their use to prevent or treat diseases and to improve the daily quality of life has led to a consumption that increases by 3% every year, so their release into the environment is also increasing [1,2].

Mebendazole is a synthetic derivative of benzimidazole and one of the best known broad-spectrum anthelmintics. It is used not only in human medicine but also in veterinary medicine, as more and more pets are infested with worms that can pose a health risk. There are a growing number of studies dealing with the use of drugs from the benzimidazole group as potential anticancer drugs for colon, ovarian, and lung cancers [3,4,5,6,7]. Studies have shown that mebendazole may be an attractive candidate for drug repositioning in metastatic colorectal cancer (mCRC) due to its efficacy in improving tumor response to chemotherapy, with studies conducted on colon cancer (CRC) patients [4]. In addition, mebendazole showed growth inhibitory effect on ovarian cancer cells [3]. Colon cancer is the third most common cancer and the second leading cause of cancer-related deaths in the world, and epithelial ovarian cancer is the deadliest gynecological cancer [3,4]. The main advantage of mebendazole as a cancer therapeutic agent is its destabilizing effect on microtubules and its safe effect compared with toxic conventional chemotherapy [3]. Although the official use of mebendazole and other benzoimidazoles in cancer therapy has not yet begun, the dissemination of information about their efficacy has led to their use as readily available nonprescription drugs [3].

Therefore, the use of mebendazole as an antiparasitic and uncontrolled use as a cancer therapy, but also according to the results of research, the undeniable future official use in the treatment of cancer, the path to an increasing presence in the environment, wastewater, natural streams and even drinking water is guaranteed. To prevent resistance to the aforementioned drug, its penetration into the environment and possible negative effects on the environment, it is necessary to develop and test methods to remove the drug from water to test its behavior under environmental conditions.

The metabolites and therapeutic effects of mebendazole have been the subject of numerous publications, while its general effects on environmental pollution and methods of its removal from the environment are poorly known. Unlike other pharmaceuticals, anthelmintics as veterinary drugs and their metabolites enter the environment directly via urine and animal faeces, through the application of contaminated manure during pasture processing and aquaculture activities [8]. Pharmaceuticals usually enter the environment through municipal, agricultural, and industrial wastewater. About 50% of anthelmintics used are discharged into public waters [9]. As a result of the massive use of anthelmintics for more than 20 years, there has been accumulation in the environment and pollution of water bodies [10]. A major problem is drug resistance, which has already been identified in nematodes for all groups of anthelmintics, and further increases in resistance are expected [11,12]. According to the World Health Organization (WHO), more than 676 million school-aged children in endemic countries were treated with anthelmintics in 2018 [13].

Membrane separation technology has become widely used in industry and wastewater treatment in recent years. Membrane treatment has been shown to reduce the incidence of contamination by antibiotic resistance genes (ARG) and other xenobiotics and to reduce the risk of their leaching into natural waters [14]. The membrane processes of reverse osmosis (RO) and nanofiltration (NF) showed excellent removal (>90%) of various pharmaceuticals [2,15]. One of the problems of RO and NF is the disposal of the retentate (concentrate), one of the output streams. The contaminants in the retentate of RO have higher concentrations than in the input stream [16,17]. It is necessary to adapt the retentate treatment technology to the source to better remove the contaminants present, which are a problem during disposal.

In this study, the removal of mebendazole by RO/NF membranes was investigated. Removal by photolysis and adsorption on granular activated carbon (GAC) was also investigated. The experiment was conducted with the aim of testing their independent application, but also as a possible treatment option for RO/NF retentate.

Adsorption is a method that is easier to use and requires fewer resources, while it has excellent efficiency in removing various xenobiotics from water. The selection of the right adsorbent and its affinity to the xenobiotics to be removed affect the working parameters and efficiency of the adsorption process. Adsorbents commonly used in water treatment include carbon materials, polymers, nanomaterials, zeolites, bioadsorbents, and waste products [18,19]. An increasing number of studies are investigating the properties and removal efficacy of xenobiotic using cheaper adsorbents such as green coffee residues, tea residues, volcanic ash, seed residues, agricultural residues, steel dust, and clays [20,21]. Although cheap adsorbents would reduce the cost of activated carbon synthesis, they have not yet been used on an industrial scale. Carbon materials are widely used in wastewater treatment to remove various contaminants. The large surface area and porous surface structure of GAC allow adsorption of organic compounds and non-polar xenobiotics, which have been shown to be effectively removed from drinking water and in the wastewater process itself [22].

Photodegradation plays an important role in the transformation of pharmaceuticals in surface waters under the influence of sunlight, and the efficiency of the process itself depends on various parameters, such as the intensity and frequency of light, the structure of the pharmaceutical, the concentrations of individual water components, etc. [23,24]. These photolytic reactions are often complex and can lead to the formation of biologically active transformation products, that may be more toxic than the parent compound and thus pose a risk to aquatic ecosystems or human health [1,9,25]. While the pharmaceutical metabolites have usually been studied, the transformation and degradation products are unknown. Therefore, it is not only important to study the removal of pharmaceuticals by photolysis, but also to determine the toxicity after this process and the resulting photodegradation/transformation products. 

To the authors’ knowledge, this is the first work to study the removal of mebendazole from an aqueous medium by RO and NF and its removal by adsorption on GAC. Moreover, for the first time, the toxicity of the studied drug in the aqueous system with *Vibrio fischeri* before and after photolytic degradation was investigated, taking into account possible transformation products that retain biological activity and reach a potentially higher toxicity than the initial component. Thus, the goals of this study were to: (i) remove mebendazole by RO and NF membranes and identify the removal mechanisms, (ii) remove mebendazole by adsorption and photolysis and investigate the influence of the initial concentration on the removal percentage, and (iii) determine the ecotoxicity of the pure mebendazole solution and the mebendazole mixture and the resulting degradation or transformation product after photolysis by *Vibrio fischeri*.

## 2. Materials and Methods

### 2.1. Chemicals

The standard of the pharmaceutical tested in this work was the anthelmintic mebendazole obtained from ACROS Organics. The purity of mebendazole was ≥97.5% according to the supplier. The chemical structure of mebendazole and its physico-chemical properties are shown in Table 1. Binary solutions of mebendazole were prepared in demineralized water. The initial concentrations of mebendazole solutions were 1 and 3 mg/L.

Yeast extract and peptone for bacterial use were purchased from Biolife (Milan, Italy), and agar for bacterial use was purchased from Liofilchem (Roseto degli Abruzzi, Italy). Other chemicals according to the standard required for toxicity testing (p.a. purities) were purchased from Lach-Ner (Neratovice, Czech Republic), Fischer Scientific (Loughborough, UK), and Thermo Scientific (Kandel, Germany).

HPLC grade acetonitrile from Fisher Scientific (Loughborough, UK) and ultrapure water (Milli-Q water) were used for liquid chromatography. Milli-Q water was prepared by the Millipore Simplicity UV-system (Millipore Corporation, MA, USA).

### 2.2. Membranes

The removal of mebendazole by nanofiltration membranes NF90 and NF270 and by reverse osmosis XLE (Dow/FilmTec, Midland MI, Michigan, USA) was investigated. The characteristics of the membranes tested have been described previously [27]. All membranes were stored in a cold, dark place until use.

The removal of mebendazole by the different membranes was performed in a laboratory set-up with six RO/NF cells connected in parallel and details have been previously published in Dolar et al. (2011) [28]. This apparatus was used to show the removal efficiency and removal mechanism of mebendazole with RO/NF membranes. Concentrating of mebendazole solutions with the membranes studied was tested in a laboratory set-up previously described by Racar et al. (2020) [29].

Prior to testing, the membranes were soaked and washed with approximately 5 L of demineralized water (without pressure) to wash out conserving agent. Then, the membranes were pretreated under a pressure of 12 bar to stabilize the permeate flux. After pretreatment, the flux was monitored at an operating pressure of 10 bar.

In the removal efficiency experiments, the pharmaceutical solution (1 and 3 mg/L) was treated at operating pressure for 2 h. Concentrating of the mebendazole solution was carried out at operating pressure using the test membranes. The volume of mebendazole feed solution was reduced from 5 to 1 L with retentate recirculation to study the dependence of retentate concentration on volume reduction factor (VRF) and to investigate the influence of feed solution concentration on permeate quality and possible membrane adsorption. Samples of permeate and retentate were taken at appropriate solution volumes in the tank. The *VRF* was calculated according to the following equation:(1)$$VRF=\frac{V\left|0\right|}{V\left|R\right|}=\frac{V\left|0\right|}{V\left|0\right|-V\left|P\right|},$$
where *V*|0| is initial volume, *V*|*R*| is retentate volume, and *V*|*P*| is permeate volume.

The concentrations of mebendazole in all samples were determined by high performance liquid chromatography with diode array detector (HPLC-DAD). After the process, the membranes were washed with approximately 30 L of demineralized water under working pressure.

After completion of the experiment, the membranes were removed from the apparatus and dried in an oven at 35 °C for 15 h. The dried membranes were used for further analysis with a Fourier transform infrared spectrophotometer (FTIR).

### 2.3. Adsorption

Column adsorption was tested on a laboratory scale using a glass column (L = 50 cm and d = 3 cm). The height of GAC poured into the column was 5, 10, 20, and 40 cm. The total volume of leaked mebendazole solution was monitored with a graduated vessel placed under the column opening. The flow rate was monitored by sampling and weighing on an analytical balance. It was necessary to ensure a continuous flow of the mebendazole solution through the column and to maintain a constant level of the solution in the column. Samples for analysis were taken before starting and effluent samples were taken at the exit of the column after 200, 400, 600, 1000, 1500, 2000, 2500, 3000, 3500, and 4000 mL of treated solution. The same procedure was repeated for each height of GAC at a given concentration of mebendazole solution. The effect of pH-values on pharmaceutical removal by adsorption on GAC was also investigated. The pH-values were adjusted with hydrochloric acid and sodium hydroxide.

### 2.4. Photolysis

Photolytic degradation of pharmaceutical in solution was studied using a Suntest CPS+ apparatus (Atlas, Linsengericht, Germany), the characteristics of which have been described in detail previously [30]. Adjustable parameters are temperature, radiation intensity, and exposure time of the radiation to the sample.

Solutions with a volume of 40 mL and concentrations of 1 and 3 mg/L were irradiated in quartz vials for 2 h, and samples were taken at appropriate time intervals and analyzed using HPLC-DAD. Solutions of 1 and 3 mg/L were also irradiated under the same conditions but for 12 h. The irradiation intensity was kept at 500 W/m^2^ and the temperature at 25 °C in both cases.

The same volume of 1 and 3 mg/L solution was illuminated in the same way, but the quartz vials were protected with aluminum foil to avoid radiation exposure. This study was conducted to determine if the decrease in concentration was due to photolytic degradation rather than hydrolytic degradation.

The photolysis rate (*r*) was calculated according to equation:(2)$$r=\frac{\left(c_{0}-c_{t}\right)}{c_{0}}\cdot 100\%,$$
where *c*_0_ is the initial concentration of mebendazole and *c_t_* is the residual concentration at time *t* [31]. By processing the data, the coefficients of mebendazole degradation rate (*k*) and half-life (*t*_1/2_) were calculated according to the following expressions:(3)$$-\ln\frac{c_{0}}{c_{t}}=k\cdot t$$
(4)$$t_{1/2}=\frac{\ln2}{k}$$

### 2.5. Toxicity

The toxicity test was performed according to standard bioluminescent method ISO 11348 with fresh *Vibrio fischeri* bacteria. Lyophilized bacteria were obtained from Hach Lange (Luminiscent bacteria test LCK 484, Düsseldorf, Germany), and a culture of the same fresh bacterium was prepared from the lyophilized bacteria on a microbial growth medium, which was then used for further measurements.

Bacterial luminescence measurements were performed using a LUMIStox 300 from Hach Lange instrument (Düsseldorf, Germany), which was used with a thermostatted LUMIStherm block to incubate the bacteria.

Luminescence was monitored initially and after 30 min as a parameter for indicating toxicity or inhibition. All measurements were performed at an instrument operating temperature of 15 ± 1 °C. The pH-value of the resuspension solution and the 2% NaCl solution was adjusted to 7.0 ± 0.2 to avoid a negative effect of pH-value on bacterial luminescence. Adjustment was made by adding sodium hydroxide or hydrochloric acid and measuring the pH-value with the S20 SevenEasy pH meter (Mettler Toledo, Greifensee, Switzerland). The geometric sequence with 2% NaCl solution was used to prepare the dilutions and measurements. The geometric sequence leads to dilution of the original sample (the last sample was diluted 128 times), which allows toxicity tests in a wide concentration range.

### 2.6. Chemical Analysis

Analysis and quantification of mebendazole in the samples was performed using a Varian ProStar 500 (Walnut Creek, CA, USA) high performance liquid chromatography system with a ProStar 330 diode array detector (DAD) (Walnut Creek, CA, USA) as previously described [32]. An InertSustain C18 250 mm × 4.6 mm column, 5 μm particle size (GL Sciences, Tokyo, Japan) was used for analysis. Chromatograms were recorded at wavelengths from 230 to 365 nm, with separation monitored at an absorbance wavelength of 254 nm.

The mobile phase used for chromatographic separation consisted of solvent A (0.01% formic acid in Milli-water) and eluent B (0.01% formic acid in acetonitrile) with gradient elution. Elution began with a 2.3-min linear gradient from 100% A to 8% B, followed by a 3.7-min linear gradient to 10% B, a 5-min linear gradient to 30% B, a 4-min linear gradient to 60% B, and finally a 3-min linear gradient to 95% B maintained for 10 min, and a 0.1-min linear gradient back to initial conditions. Each sample was injected three times, and the injection volume was 30 µL. The flow rate was 0.5 mL/min. The pressure in the system ranged from 20 to 25 atm and the temperature from 20 to 25 °C. The retention time of mebendazole was 22.907 min. The limit of detection (LOD) in demineralized water was 0.1 mg/L and the limit of quantification (LOQ) was 0.3 mg/L. The calibration curve was linear with R^2^ > 0.9906.

The photolytic test samples were also analyzed using Shimadzu LC-20ADXR high performance liquid chromatograph (Shimadzu, Kyoto, Japan) with SPD-20AV UV detector and LCMS 2020 mass spectrometer (Shimadzu, Kyoto, Japan). The analysis was performed under the same conditions as for HPLC-DAD.

### 2.7. Fourier Transformation Infrared Spectrometer—FTIR

Possible interactions between membranes and pharmaceutical were investigated using FTIR (Bruker Vertex 70, Ettlingen, Germany) equipped with a Platinum ATR single reflection diamond (*n* = 2.4) crystal-based module in the middle IR range (400–4000 L/cm). Prior to scans of the membrane surface samples (≈0.5 × 0.5 cm), the background spectrum (air) was recorded to avoid measurement errors. The scans were performed before and after pharmaceuticals tests.

## 3. Results and Discussion

### 3.1. Removal of Mebendazole with RO/NF Membranes

XLE, NF90, and NF270 were selected as typical RO, tight NF, and loose NF membranes, respectively, to determine the removal rate and removal mechanisms of mebendazole. The rejection factors of mebendazole in RO/NF treatment are shown in Table 2.

As can be seen, the highest removal was obtained with XLE and NF90 membranes, followed by NF270 membrane. The RO XLE and tight NF90 membranes showed almost complete removal for all investigated concentrations. From Table 2, it can be seen that these membranes have the best and quite similar retention factors of more than 97.7%. Considering the possibility of predicting the rejection factors based on molecular weight, a higher rejection factor is expected for XLE and NF90 membranes and a lower one for NF270. The MWCO for XLE membrane is 100 Da and for NF90 is 100–200 Da [32]. Therefore, the molecular weight of mebendazole (Table 1) is higher than the MWCO value. The MCWO value of 150–300 Da of NF270 membrane could explain the lower rejections [32].

The size exclusion mechanism was also confirmed by comparing the pore sizes of the membranes and the dimensions of the components shown in Table 1. It was found that the pores of the membrane RO XLE membrane were centered at 0.33 nm, while for NF270 they were 0.42, 0.64, and 1.36 nm [33]. NF90 as tight nanofiltration membrane is very similar to RO membrane [28,34]. From the dipole moment and the structure of the molecule, it can be concluded that the methyl group enters the membrane structure at the end of the molecule, which means that the height and width are the dominant parameters. These are the reasons for the lower removal of mebendazole with the NF270 membrane and the almost complete removal with the XLE and NF90 membranes. From Table 2, it can be seen that the initial concentration of the compounds in the feed solution is an important factor affecting the rejection efficiency [35]. For the XLE and NF90 membranes, the rejection decreased between 1.6 and 2.3% for 5 and 10 mg/L solutions due to higher concentration polarization and probable adsorption. However, for the NF270 membrane, an increase in concentration resulted in an increase in the rejection factor from 41.9% to 96.6%. It can be assumed that adsorption and hydrophobic interactions lead to an increase in rejection. Adsorption on the membrane surface or inside the pores narrowed the pores and decreased the transport of pharmaceuticals through the membrane. The high adsorption of mebendazole on membranes is favored by its low water solubility (71.3 mg/L) and high log*K*_O/W_ value (2.83). The adsorption of organic compounds increases with decreasing water solubility, and components with a log*K*_O/W_ value above 2 are considered hydrophobic and are likely to adsorb on the RO/NF membranes [35,36]. Mebendazole may associate with the -CH_3_ functional group, which may increase the effect of size exclusion and adsorption of pharmaceuticals on the outer surface or in the pores of the membrane. FTIR analysis was performed to determine the possible adsorption of mebendazole on membranes. The results showed that there were no significant changes in the structure of the XLE membrane. As shown in Appendix A, a number of new peaks were observed for the NF90 and NF270 membranes. The new peaks for the NF90 membrane are visible at 769, 907, 977, 1640, 1732, and 3371 cm^−1^, and for NF270 at 772, 906, 1530, 1640, 1730, and 3371 cm^−1^. For the XLE membrane, only one new peak (Appendix A) is visible at 1732 cm^−1^, which represents a new carbonyl group [33]. Peaks in the range between 1750 and 1700 cm^−1^ are generally simple carbonyl compounds such as ketones, aldehydes, esters, or carboxyl [37]. The changes in NF90 and NF270 membranes are similar and can be explained by the pore size, due to which the possibility of mebendazole penetration into the membrane pores is higher, resulting in a significantly higher number of new peaks in the FTIR spectrum. Peaks at 3371 cm^−1^ were observed in NF270 membrane and NF90, and according to the literature, the absorption band in the range between 3650 and 3250 cm^−1^ indicates hydrogen bond [33,37]. The peak at wavenumber 1640 cm^−1^, reflecting functional groups of primary and secondary amides, could be due to H-bonding and vibrations of amide C=O bonds or alkenyl C=C stretching [33,37,38]. The region 800–1000 cm^−1^ showed the string bending of methyl (-CH_3_) group and at 772 cm^−1^ the bending of -CH_2_ [33]. 

From the rejection factors and FTIR results, it can be concluded that size exclusion prevails as a separation mechanism for RO and tight NF membranes, while physico-chemical interactions had the greatest influence for loose NF membranes. 

In further work, the selected membranes were used in a larger apparatus (Sepa II) with better hydrodynamic conditions and a larger surface area to investigate the effect of volume reduction on mebendazole concentration in the retentate. The mebendazole concentration in the retentate compared to the VRF is shown in Figure 1. 

As described earlier, adsorption on the membrane is expected due to the hydrophobicity, dimensions, dipole moment, and low solubility of the molecule. This effect is again most evident with the NF270 membrane. Due to reduction in volume during retentate recirculation, an increase in concentration in the feed solution is expected unless mebendazole is adsorbed on membranes or parts of the apparatus. However, for the XLE and NF90 membranes, there was an increase in the concentration in the feed solution, whereas for the NF270 membrane, the concentration in the feed began to decrease at VRF 1.66 and increased at VRF 5. The unexpected behavior of the curve for the NF270 membrane (Figure 1) can be related to the weaker removal, larger pores of the mentioned membrane, and the adsorption of mebendazole on the surface and in the pores of the membrane which is expected due to the size of the pores and the molecule and was also detected by FTIR. Greater adsorption at the surface and in the pores leads to a decrease in mebendazole concentration in the recirculating retentate stream, but also to weaker removal. As expected, a higher concentration of mebendazole on the membrane surface and in the pores leads to a higher penetration of mebendazole into the permeate. Using a solution with a concentration of 1 mg/L as a example, the initial removal was 79.5% for *VRF* 1 and then decreases to 40.8% for *VRF* 2.5. It is assumed that after this point there is desorption of mebendazole from the surface, but also weaker passage of mebendazole through the already partially filled pores of the membrane, leading to a jump in the concentration of mebendazole in the retentate for *VRF* 5, but also to better removal, which increases to 50.2%. Adsorption and desorption of mebendazole from parts of the device can lead to a decrease or increase in the concentration in the input solution.

In addition, adsorption on the membranes leads to a decrease in the flux, which was also observed during the treatment of mebendazole solution. Throughout the solution treatment process, the permeate flux decreases, eventually decreasing from 70 to 62 L/(m^2^ h) for the XLE membrane, from 70 to 63 L/(m^2^ h) for the NF90 membrane, and from 190 to 174 L/(m^2^ h) for the NF270 membrane. The decrease in flux is attributed to adsorption in the membrane pores, which is consistent with the assumptions made previously.

Mebendazole was successfully removed from the binary solution using the membrane procedure. However, the next step of our work addresses the issue of removal in real wastewater matrices where a large number of pharmaceuticals, active components, and other organic pollutants are present. According to the available literature, higher removal can be expected especially for NF membranes due to the occurrence of synergistic effects [32,39,40].

### 3.2. Adsorption

The influence of GAC height, flow rate, pH-value, and input concentration on the removal of the tested pharmaceutical was investigated by column adsorption. The input concentration is an important factor that can affect the height of GAC applied in the column, while the layer height mainly affects the adsorption efficiency [41]. The flow rate affects the contact time between the adsorbate and the adsorbent, which significantly affects the removal efficiency [41]. The efficiency of the adsorption process is represented by the breakthrough curve.

Preliminary column adsorption test was performed with a continuous flow of mebendazole solution (3 mg/L) at a height of 5 cm GAC and a flow rate of about 130 mL/min. Lin et al. and Rizzo et al. have shown that the pH-value of the solution is an important parameter for controlling the adsorption process. Therefore, tests were performed at pH-values of 4.12, 6.88 and 9.13 (Figure 2) [42,43].

Due to the high flow rate and the low height of the GAC (5 cm), i.e., due to the very short contact between the solution and the adsorbent, there was no significant decrease in the mebendazole concentration. The largest decrease in concentration was observed at neutral pH-value (18.0% ± 2.8). The breakthrough curves are shown in Figure 2, and it can be seen that the saturation of the activated carbon and a higher concentration of the effluent occur in the acidic pH range. This is probably due to the poorer adsorption and easier desorption in the indicated pH range as a result of the weak adsorption bonding energy [43]. 

According to the results obtained previously, the work was continued at neutral pH-value, lower flow rate (30 mL/min) to ensure better and longer contact between the adsorbent and the adsorbate, and at different height of the GAC column to ensure more adsorption sites. Compared with the previous adsorption at different pH-values (Figure 2), where the flow rate was about 130 mL/min, it can be concluded that the adsorbate has more time to diffuse into the pores at lower flow rate, resulting in higher adsorption, i.e., greater removal of mebendazole. To further reduce the flow due to a significant slowdown of the process, the tests were continued with different amounts of GAC, i.e., GAC height (Figure 3). 

At higher GAC heights, a greater number of adsorption sites are available, ensuring greater efficiency in mebendazole removal, but also slower saturation of the GAC compared to lower heights. At the beginning of the adsorption process, the mebendazole concentration decreases sharply and the process continues with a moderate increase, which is visible on the breakthrough curve (Figure 3) by increasing the *γ/γ*_0_ ratio. Each adsorbent has a number of sites on its surface to which adsorbate molecules can bond. They fill quickly at first, but as occupancy increases, the probability of the adsorbate molecule coming into contact with the free binding site decreases. The greatest decrease in mebendazole concentration occurred with the adsorption column containing the highest amount of activated carbon.

To test the influence of concentration, adsorption was performed with a continuous flow of mebendazole solution (1 mg/L) at a height of 40 cm GAC and a flow rate of about 30 mL/min in the neutral pH range. The effluent concentrations in all samples collected were below the detection limit (results not shown), and it can be concluded that the removal of mebendazole from the 1 mg/L solution at a height of 40 cm GAC was complete, as shown in the chromatogram (Appendix A). It can be concluded that a higher concentration means a larger amount of adsorbate coming into contact with the adsorbent before the GAC is saturated, or more occupied adsorption sites, resulting in poorer removal at higher concentrations.

### 3.3. Photolysis and Toxicity

Direct photolytic degradation was tested in 1 and 3 mg/L mebendazole solution prepared in ultra-pure water. Photolysis was performed for 2 and 12 h. All 12 h photolysis experiments were performed in triplicate, and samples were analyzed using a HPLC-DAD device. Due to the chemical structure of mebendazole, photosensitivity is expected [44]. 

There was no decrease in concentration over time in the control samples protected with aluminum foil, indicating that photolysis during the test was mainly caused by radiation. The results of the photodegradation experiment of mebendazole showed that it was susceptible to degradation, with the highest photolytic degradation rate being 43% after a 1 mg/L solution was exposed to simulated solar radiation for 120 min. The photolysis rate was calculated using Equation (2). As shown in Figure 4, the photolytic degradation rate is lower (37%) in a solution with a 3-mg/L concentration.

There were no significant changes in the photolytic degradation rate during the 12 h photolysis (Figure 4). 

In the experiments, linear regression ln(*c*_0_/*c*_t_) with respect to time (*t*) showed that photodegradation followed first-order kinetics with *R*^2^ > 0.93, as expected in the literature [44]. By processing the data, the coefficients of the rate of degradation of mebendazole (*k*) were calculated, i.e., the half-life was calculated using Equation (4) (*t*_1/2_), which is 12 h for a concentration of 1 mg/L and 28 h for a solution of 3 mg/L.

The photodegradation durations of 2 and 12 h were not sufficient to completely degrade mebendazole. However, in both experiments, a new component is formed during photolytic degradation of mebendazole, which is visible on the chromatogram in Appendix A. The new component appears at a retention time of 23.467 min and has a maximum at about 203, 253, 312 nm, while the retention time of mebendazole is 22.940 min and the maxima are at about 210, 246, 309 nm, as shown in Appendix A.

Analysis of the samples after photolysis on the LC-MS device confirmed that the mebendazole molecule is *m/z* 295 and the degradation or transformation product is *m/z* 311 (Appendix A). It is suggested that the entry of the OH group into the aromatic core of the ring (Figure 5) occurred under the influence of photolytic radiation in the aqueous medium.

In accordance with the subject of the work, the structure was not further investigated because it requires the synthesis of the newly formed compound and further analytical studies.

Toxicity analyses should be performed for the parent component, but also for the degradation/transformation products after photolysis, as other, more toxic products may be formed. Toxicity analyses using a luminescent bacterial test have shown that UV radiation reduces the toxicity of some pharmaceuticals to aquatic organisms by reducing their amount, but also increases the toxicity of other pharmaceuticals, producing toxic products [45]. For these reasons, a toxicity study of the parent component with *Vibrio fischeri* was performed, but also a test after 12 h of photolysis. Toxicity was also tested for samples after 4 h of photolysis, after a preliminary study showed that the highest concentration of the unknown component was present at this time.

Toxicity evaluation results for samples after photolytic degradation of mebendazole at initial concentrations of 1 mg/L and 3 mg/L are shown in Table 3.

The inhibition of bacterial growth after 30 min of exposure to the standard solution is at a very low percentage, indicating very low toxicity of the mebendazole solution. Inhibition increases more weakly when the concentration of the degradation or transformation product is highest, but this increase is not significant. After 12 h, inhibition no longer occurs, which can be explained by lower concentrations of mebendazole, but also of the degradation product in the sample. In addition to the ecotoxicity analysis, it is necessary to identify the resulting degradation or transformation product to better follow its fate and behavior in the environment.

The toxicity of mebendazole has been studied exclusively in dogs, sheep, rabbits, rats, and other rodents [46,47]. In these studies, fetal toxicity and teratogenicity in rats, anemia, and increased levels of alkaline phosphates, bilirubin, cholesterol, and total serum proteins as a result of a high-dose of mebendazole have been reported, and testicular damage was demonstrated. It is clear that the ecotoxicity, behavior, and fate of mebendazole in the environment have not been adequately studied.

## 4. Conclusions

The aim of this study was to investigate the behavior and removal of mebendazole from aqueous media by RO/NF membranes and its removal from retentate by adsorption and photolysis. The dense NF90 and RO XLE membranes showed almost complete removal (>97.7%), while in the case of the loose NF270 membrane (41.9–96.6%), the initial concentration had a high influence.

GAC successfully removed mebendazole from the retentate, and the flow rate, the height of the carbon layer, and the initial concentration of the solution had a great influence on the efficiency. 

Photolysis is also a suitable method to remove mebendazole from retentate, and it can be concluded that mebendazole is a photosensitive pharmaceutical. However, a negative aspect of photolysis was the formation of photodegradation product.

## Figures and Tables

**Figure 1 membranes-12-00888-f001:**
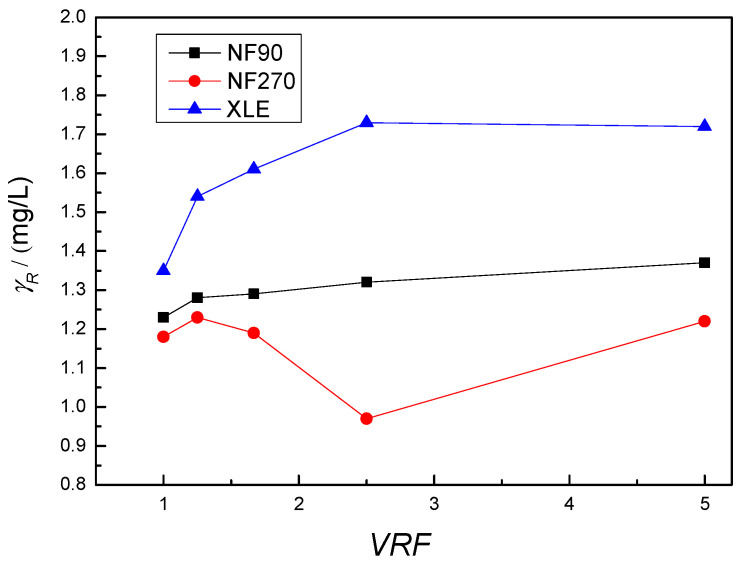
Dependence of mebendazole concentration in retentate on *VRF* value.

**Figure 2 membranes-12-00888-f002:**
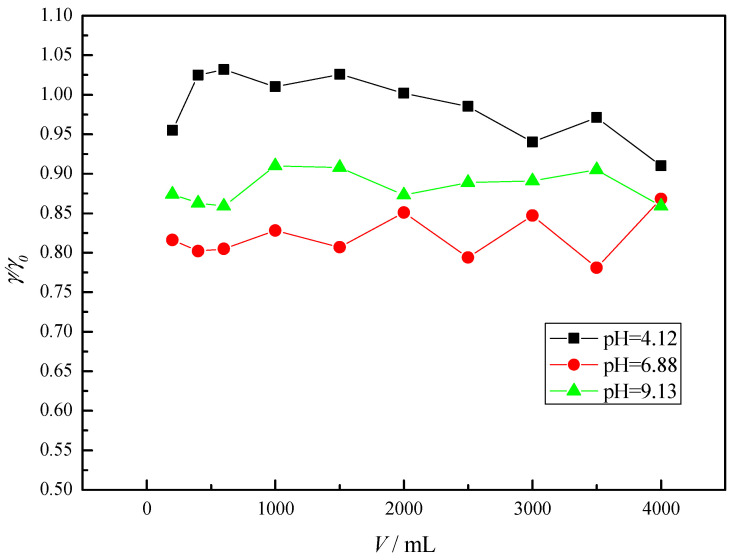
Breakthrough curves for mebendazole solutions at different pH-values.

**Figure 3 membranes-12-00888-f003:**
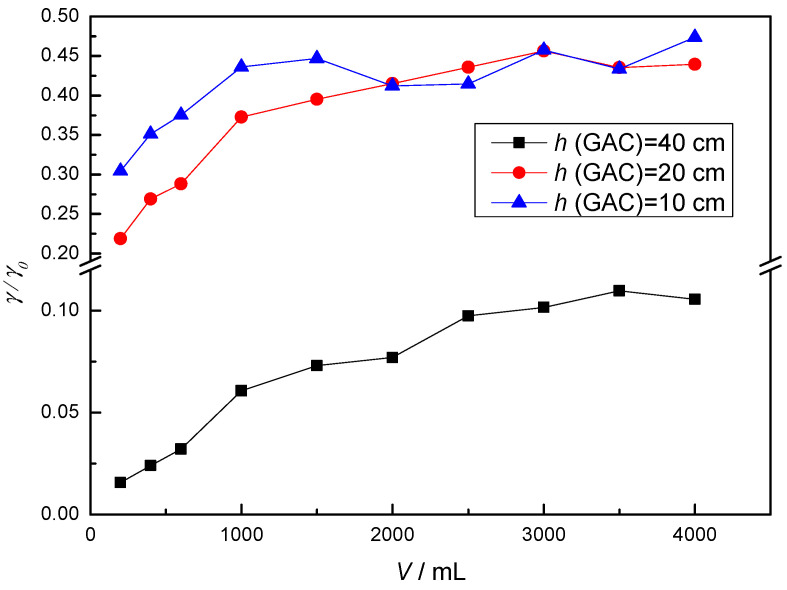
Breakthrough curves at different GAC column heights.

**Figure 4 membranes-12-00888-f004:**
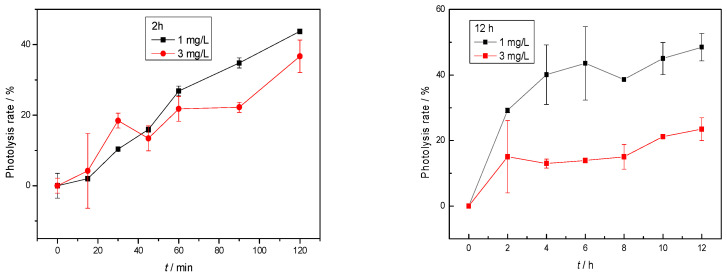
Removal rate of mebendazole during 2 h and 12 h of photolysis. Error bars represent the standard deviations (*n* = 3).

**Figure 5 membranes-12-00888-f005:**
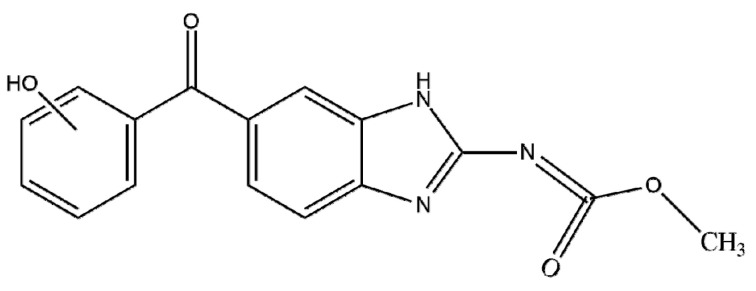
Proposed structure of the newly formed compound.

**Table 1 membranes-12-00888-t001:** Physico-chemical and general properties of mebendazole.

IUPAC Name	Methyl *N*-(6-benzoyl-1H-benzimidazol-2-yl)carbamate
CAS number	31431-39-7
*M*_w_ (g/mol)	295.293
Molecular structures	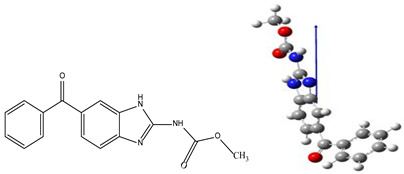
width (nm) ^b^	0.511 ^b^
height (nm) ^b^	0.418 ^b^
length (nm) ^b^	1.572 ^b^
log*K*_ow_	2.83 ^a^
*t*_t_ (°C)	223.41 ^a^
*t*_v_ (°C)	523.14 ^a^
*S* (mg/L)	71.3 at 25 °C ^a^
Dipole moment, *µ* (D)	6.318 ^c^

^a^—EPISuite; ^b^—calculated using the HyperChem program; ^c^—Dipole moment calculated by Gaussian [26].

**Table 2 membranes-12-00888-t002:** Mebendazole concentrations in permeate and rejection factors for NF270, NF90, and XLE membranes.

	XLE	NF90	NF270
	1 mg/L		
*γ*/(mg/L)	<LOD	<LOD	0.60
*R*/%	>99.9	>99.9	41.9
	3 mg/L		
*γ/*(mg/L)	<LOD	<LOD	1.29
*R/%*	>99.9	>99.9	57.0
	5 mg/L		
*γ/*(mg/L)	0.11	0.10	0.33
*R/%*	97.7	97.9	93.2
	10 mg/L		
*γ/*(mg/L)	0.17	0.17	0.36
*R/%*	98.4	98.4	96.6

**Table 3 membranes-12-00888-t003:** Inhibition of samples before and after photolysis.

	1 mg/L		
*t*/h	0	4	12
INH/%	4.01	10.25	0
	3 mg/L		
*t*/h	0	4	12
INH/%	6.66	9.71	0

## Data Availability

The data presented in this study are available on request from the corresponding author.

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
