# Peer review of "Behavior of Mebendazole during NF/RO Adsorption and Photolysis"

_membranes, 2022, doi:10.3390/membranes12090888_

Round 1
Reviewer 1 Report
Dear Authors
The paper itself is well written, although somewhat descriptive. You all have conducted a thorough literature review, undertaken a rigorous piece of data collection and have analyze information accurately.
Thanks

Author Response
Reply to all comments in in the document. Hope we answer to all comments successfully.

Reviewer 2 Report
Please address the following comments:
Activated carbon will be required to remove chlorine etc before RO, otherwise it will reduce the life of the membrane?
I wonder why we need to use RO instead of AC? please justify in relation to operation and economic?
There is may be thousand of pharmaceutical chemicals in the water, how the membrane will respond to this?
Can you present the flux and the removal vs time for NF90, NF270 and XLE?
Please incorporate your answers into the article.
Author Response

(The authors gave the same response as above.)

Reviewer 3 Report
This paper is good, but it should be improved.
- Figure 1 shows the dependence of the substance concentration in the retentate depending on the concentration factor. The line shape for the NF270 membrane (red line) is very astounding. The authors perceived this as well but made no attempts to explain such an unusual shape. I am looking forward to an in-depth discussion.
- the authors study adsorption on membranes and on GAC. There is no discussion of what should be done with membranes and GAC after their lifetime so that they do not become a source of recontamination.
Author Response

(The authors gave the same response as above.)

Reviewer 4 Report
Nice work but its needs moderate revision.
1. Add the purity of the used chemicals.
2. Improve the conclusion part.
3. The authors should propose the mechanism for the removal of mebendazole.
4. In fig. 4: The authors need to check other ppm levels.
5. The authors should compare the present removal mebendazole results with reported previously published results.
6. The authors should include the stability analysis after 2 hrs of photolysis.
7. More typo errors in the manuscript. Check it very carefully.
Author Response

(The authors gave the same response as above.)

Round 2
Reviewer 3 Report
I am satisfied with the correction. The paper is ready to publish.
Author Response
Thank you for being satiesfied with the changes. English was checked again. We hope now is ok.